# Evaluating Breastfeeding Self-Efficacy in Adolescents Attending a Co-Designed Breastfeeding Program: A Prospective Pilot Cohort Study

**DOI:** 10.3390/ijerph22081271

**Published:** 2025-08-14

**Authors:** Christina M. Cantin, Wendy E. Peterson, Amisha Agarwal, Jemila S. Hamid, Bianca Stortini, Nathalie Fleming

**Affiliations:** 1Children’s Hospital of Eastern Ontario Research Institute, Ottawa, ON K1H 8L1, Canada; 2Faculty of Health Sciences, School of Nursing, University of Ottawa, Ottawa, ON K1S 5S9, Canada; 3Employment and Social Development Canada, Ottawa, ON K1A 0J9, Canada; 4Department of Mathematics and Statistics, University of Ottawa, Ottawa, ON K1N 6N5, Canada; 5Department of Obstetrics and Gynecology, Faculty of Medicine and Health Sciences, University of Sherbrooke, Sherbrooke, QC J1K 2R1, Canada; 6Faculty of Medicine, University of Ottawa, Ottawa, ON K1H 8M5, Canada; 7Pediatric and Adolescent Gynecology, Children’s Hospital of Eastern Ontario, Ottawa, ON K1H 8L1, Canada; 8Department of Obstetrics/Gynecology, The Ottawa Hospital, Ottawa, ON K1H 8L6, Canada

**Keywords:** lactation, breastfeeding, self-efficacy, peer support, adolescent, community-based intervention program

## Abstract

Adolescents have lower rates of breastfeeding (BF) compared to older mothers. BF self-efficacy (SE) has been identified as an important factor influencing BF outcomes. An innovative BF program for young women was co-designed and implemented, which included staff training, a prenatal BF class, and BF peer support. The objective of this prospective pilot cohort study was to evaluate the effectiveness of prenatal education and peer support in improving a young mother’s BF SE. Participants were pregnant adolescents recruited from a large urban non-profit social service outreach centre. The Breastfeeding Self-Efficacy Scale-Short Form (BSES-SF) was administered to participants before and after participating in the BF program. BSES-SF scores were summed to determine a composite score and compared descriptively using median score. Un-aggregated, item-by-item, comparison of pre- versus post-BF program scores were also compared to examine improvements in SE. A total of 20 adolescent mothers (mean age = 16.6) attended the program. An increase in the total BSES-SF score was observed based on descriptively comparing the mean pre- versus post-intervention. Participation in tailored prenatal education classes and a peer-support program was associated with increased BSES-SF. Identifying mothers with low BF SE can enable healthcare professionals to implement targeted interventions in this at-risk population.

## 1. Introduction

Adolescents, defined as women less than 20 years of age, have a lower rate of breast-feeding (BF), and traditional BF support given to older mothers may not meet the needs of pregnant and parenting youth [1,2,3]. Given that the benefits of BF are well known, the World Health Organization (WHO) recommends exclusive BF up to 6 months of age, with continued BF along with complementary foods up to 2 years of age or beyond [4]. This feeding recommendation is consistent with Canada’s nutrition guidelines for healthy term infants [5]. Most women in Canada initiate BF (91%). However, only 69% of women continue to breastfeed their infants exclusively for 6 months [6]. The low rate of exclusive BF exemplifies the need for targeted BF interventions to enhance BF rates.

Public health reports and studies have identified that adolescent mothers (15–19 years) have lower rates of BF initiation, BF at time of hospital discharge, and BF continuation compared to adult women [1,2,3,7]. Compared to adults, Ontario adolescents have a 27% lower rate of exclusive BF at hospital discharge [8]. Furthermore, a large retrospective population-based cohort study in Ontario (N = 22,023) revealed that only half (48.8%) of adolescent women less than 20 years of age exclusively breastfed their infant at time of hospital discharge [9]. These findings reiterate the WHO’s identification of adolescent mothers as a group of individuals facing difficult situations who require special attention and practical support [10]. Higher rates of initiation and continuation have been reported when young women received BF education and counselling [3,8,11].

Adolescent mothers encounter multiple challenges, which contribute to the low rates of intention, initiation, and duration of BF. These challenges include the following: limited access to the social determinants of health (poverty, food, and housing insecurity); negative effect on maternal confidence and attitudes by lower educational levels; low BF intention/confidence; possible negative attitude towards BF; high-risk behaviours in pregnancy (smoking and substance use); higher rates of mood disorders and violence in pregnancy; and lack of accessible support and adolescent friendly services [3,12,13].

Adolescents are also faced with simultaneously integrating new parental roles and romantic relationships, managing struggles with self-esteem and self-image, and grieving the loss of former peer groups [2]. Tailored pre- and postnatal care may have a pivotal role in improving BF education, intention, initiation, and duration in this population [3,8,9,10,11]. This education and support, given by laypeople or professionals, is effective in increasing BF duration in high-income countries [2]. Indeed, at the time of our program development, BF peer support was identified as a promising strategy to reach and support populations, such as young mothers, with lower rates of BF [1,14]. Peer support provides a low-cost, non-medical intervention by caring, passionate, and non-judgmental women with BF experience [15].

BF self-efficacy (SE), defined as the mother’s confidence in her ability to breastfeed her infant [16], has been found to be an important factor associated with BF duration and exclusivity. Studies of adolescent mothers demonstrate that BF self-efficacy (SE), both pre- and postnatally, is an important factor influencing BF outcomes [16,17]. Adolescents with low postnatal BF SE are nearly 4 times more likely to quit BF prior to 28 days postpartum [17].

Informed by these research findings and guided by our own experience working with adolescents [18], we used a participatory approach to co-design a new BF Program to address a gap in BF support for young pregnant and parenting youth in 2015 [19]. We conducted focus groups with pregnant and parenting youth (N = 16) resulting in recommendations for a new program, which included staff training, a revision of prenatal BF class curriculum, and the creation of a new BF peer-support program [19]. The program continues to be offered today.

We report the findings of a pilot study evaluating the effectiveness of prenatal education and peer support in improving BF SE. Our hypothesis was that adolescent mothers would demonstrate increased BF SE after attending one prenatal BF class and/or four BF peer-support sessions.

## 2. Methods

A prospective pilot cohort design was used, and a survey was administered before and after the intervention. Ethical approval for research involving humans was obtained from the University of Ottawa Research Ethics Board (#H02-15-09B).

### 2.1. Setting

The study was conducted in a Canadian non-profit social service agency that provides a full range of services for young pregnant women, young mothers and fathers (<25 years), and their children (<3 years of age) [20]. This agency supports young women to have a healthy pregnancy and birth and helps young parents gain the knowledge and skills to raise healthy and happy children. The outreach centre offers a one-stop location for health and social programming and services. In general, women attending programs at this agency have low-income, low educational attainment, variable family support, and housing instability. Consequently, most clients receive social assistance and rely on supports such as the food bank and the clothing bin. At the time of the BF program development and until mid-2024, there was a residence for up to 10 pregnant and five postpartum clients and their babies, which provided stable housing for this often precariously housed population.

### 2.2. Participants

The targeted study participants were adolescents (19 years of age or less) currently registered as clients at the agency described above. The following inclusion criteria was used: (1) currently pregnant or parenting, (2) attending a prenatal BF class or the BF peer-support program, and (3) could read and write in English. Participants older than 19 years of age were excluded from our analysis. Participants were recruited at the beginning of the prenatal BF class or at the beginning of the first BF peer-support session they attended. Recruitment was facilitated by verbal invitation by agency staff at selected programs and by posting flyers in the common areas of the agency. Participants were eligible to participate in the study if they met the inclusion criteria.

### 2.3. Intervention

The new program included a revision of prenatal BF class curriculum, the creation of a new BF peer-support program, and staff training. The prenatal BF class was one session in a series of eight prenatal classes on various topics (the name of the program is *Pregnancy Circle*). The 2 h class was facilitated by a public health nurse from a local public health agency, which received the Baby-Friendly Initiative Designation in May 2013, and thus the curriculum was consistent with the Baby Friendly Initiative Standards [21,22]. This class was available to pregnant youth and their partners or support persons. The curriculum was revised to reflect the realities of young mothers, such as their single status, their attendance at school versus employment, and common social situations that would be relevant to youth.

The weekly BF peer-support sessions (1.5 h) were co-facilitated by a program coordinator from the agency where the study was being conducted and a youth leader who received peer-mom leader training (the name of the program is *Little Milk Miracles*). There were four sessions that covered a range of topics under the following themes: getting off to a good start, BF in the early days, troubleshooting common concerns, and continuing to BF. Childcare, transportation, and a healthy snack were provided as this facilitated young mothers’ participation in the program. Prenatal or postnatal youth could attend the peer-support sessions.

Peer-mom leader training for the Little Milk Miracles program consisted of four hours of training, which was delivered over two sessions by the program coordinator and a public health nurse who was an International Board Certified Lactation Consultant (IBCLC). The training consisted of interactive discussions and case studies, which enabled participants to review basic BF information and to better understand the role and expectations of peer-mom leaders as well as learn about available community resources. Peer-mom leaders were selected based on having 4 months or more breastfeeding experience, possessing qualities conducive to the role (i.e., approachable, positive, supportive), and a satisfactory police record check.

The staff training consisted of a two-hour session focused on evidence-based BF support for young mothers. A public health nurse, who was an IBCLC, provided this training, which included an oral presentation, interactive discussion, and case studies. This was offered on two separate days to allow all staff at the agency to participate in the training, regardless of their role (e.g., receptionist, administrators, program coordinators, residence staff, etc.). The purpose of the staff training was to foster a culture of breastfeeding support at the agency. This component of the BF program was not evaluated as part of the prospective pilot cohort study.

### 2.4. Measurements

To measure BF SE, we developed a pre and post questionnaire. The pre questionnaire included twelve demographic questions: age, marital status, employment status, obstetrical history, parenting status, current pregnancy status (if pregnant, # of weeks), BF education received, previous BF experience (yes/no, if yes for what duration), concerns related to BF, awareness/knowledge of community resources, how they heard about the program, and reason they came to the program.

The Breastfeeding Self-Efficacy Scale-Short Form (BSES-SF) was used in both the pre and post questionnaire, as it has been validated for use with adolescent mothers [16,17]. The BSES-SF is a 14-item self-report instrument measuring BF confidence. Each item is a 5-point Likert-type scale question, where 1 indicates “not at all confident” and 5 indicates “always confident”. Possible scores ranged from a minimum of 14 to a maximum of 70 [16]. A BSES-SF cut-off score of 52 or greater demonstrates high confidence. Determining BSES-SF scores is important, because it predicts BF initiation, duration, and exclusivity [16,17].

### 2.5. Data Collection

Data collection occurred between November 2015 and October 2016. Participants who consented to participate in the study were recruited from the Pregnancy Circle prenatal classes and/or the Little Milk Miracles peer-support program. Prenatal BF Class: Participants completed the pre-intervention questionnaire prior to attending the BF class and they completed the post-intervention questionnaire at the end of the two-hour class. BF Peer Support: Participants completed the pre-interventions questionnaire at the beginning of the first BF peer-support session they attended, and they completed the post-intervention questionnaire after attending four sessions. Participants were given approximately 15 min to complete the paper-based surveys.

### 2.6. Statistical Analysis

Demographic variables for the study participants were summarized using descriptive statistics, where mean and standard deviation (SD) or median and interquartile range (IQR) were used for continuous outcomes. Frequency and percentages were used for categorical variables. Important concerns about breastfeeding and reasons for attending the program were summarized in a similar manner. For all participants with complete data (i.e., all questionnaire items answered pre and post), total scores were aggregated by summing values across all questions and mean composite scores were calculated for both pre- and post-intervention data. This was done at an overall level with data from both the prenatal and peer-support group combined; we also performed analysis for each intervention group (prenatal class and peer-support group) separately. Comparisons between pre- and post-intervention measurements were made descriptively, by comparing median values between the groups. Pre- and post-intervention scores for each question were also analyzed descriptively and visually to evaluate improvements in selected items related to young mothers’ BF confidence after attending the BF program. All statistical analyses were conducted in the R statistical software, version 3.4.1 [23].

## 3. Results

Full demographic data are provided in Table 1. In total, 20 adolescents 19 years of age and under participated in the study, of whom 90% (n = 18) said they were pregnant. The mean age of participants was 16.6 (SD = 1.3) years, and most reported that they had a partner/boyfriend (n = 12; 60%). Most of the participants had never been pregnant before (n = 11; 57.9%) and therefore few had BF experience (n = 2; 10.5%).

The participants identified a variety of concerns (Table 2), which included the following: BF in public (n = 7; 35%), being unable to start BF (n = 6; 30%), not knowing what is normal with respect to BF (n = 6; 30%), and not having enough information about BF (n = 6; 30%). Not surprisingly, most participants identified wanting to learn more about BF (n = 17; 85%) as their main reason for attending the program (Table 2).

Complete pre/post pairs of data were available for only 7 of the participants (attending either the prenatal class and/or four sessions of the peer-support group). The overall composite BFSE-SF median and IQR for pre score was 50.0 (40.5, 51.5) and for the post score was 54.0 (51.5, 57.0), indicating an improvement overall (Table 3). There was only one participant attending the Little Milk Miracles mom-to-mom peer-support program who provided scores for all the questions. The pre- and post-intervention score for this individual was 54 and 55, respectively. Statistical comparisons were not made due to the small sample size.

Scores for individual BSES-SF items before and after the BF program are presented in Figure 1 and Figure 2 (n = 7 with complete pre/post data). Generally, the shifting of the color bars to right post-intervention indicates a higher proportion of participants that are confident (or very confident). For example, the second graph in Figure 1, “I think that I can always determine that my baby is getting enough”, indicates that pre-intervention, 29% of the study population were “not at all/ not very confident” and 29% were “confident/very confident”. Post-intervention, these proportions increased; 57% were “confident/very confident”. Data were also collected on clients up to 24 years old given that this is the population the agency serves. The same analysis was done including data from these individuals (N = 32), and the results were consistent with participants 19 years and younger.

## 4. Discussion

This prospective pilot cohort study revealed that adolescent mothers in Eastern Ontario, Canada, had multiple concerns, including BF in public, being unable to start BF, not knowing what is normal, and not having enough information about BF. Most participants identified wanting more information about BF as the reason for coming to the program.

The overall pre versus post comparison of composite Breastfeeding Self-Efficacy Scale-Short Form (BSES-SF) scores indicates that there was an improvement, from low to high, in the young mothers’ SE after attending the program. This preliminary finding indicated that the BF program improved the mothers’ knowledge, confidence, and satisfaction in BF. The results of our pilot study were promising given that clients attending programs at this agency had low-income requiring social assistance and food bank usage, low educational attainment, variable family support, and housing instability, all of which represented risk factors for not BF. Previous qualitative research conducted by members of our research team identified that on-going emotional and practical supports from professionals and peer supports enabled adolescent and young mothers to achieve their BF goals [24].

The only item on the BSES-SF scale that did not demonstrate an overall improvement was BF without using formula as a supplement. This is an area that warrants further investigation, as it is not known whether participants intended to combination feed or had encountered challenges resulting in supplementation with formula. In a systematic review and meta-analysis of interventions to increase exclusive BF rates in adolescents in high income countries (N = 9 studies), Buckland et al. [25] identified that high rates of formula supplementation and attrition have limited researchers’ ability to detect statistically significant effects. Regardless, peer counselling was identified as the most promising intervention to increase exclusive BF rates in adolescents [25].

In a qualitative study of adolescent mothers (aged 15–19) (N = 16), Nesbitt et al. [13] identified five influences on continued BF: (1) the impact of BF on social and intimate relationships; (2) the availability of social support; (3) the physical demands of BF; (4) mothers’ knowledge of BF practices and benefits; and (5) mothers’ perceived sense of comfort in BF. The authors suggest that professionals can engage with adolescent mothers in both the prenatal and postnatal period to assess potential barriers to BF, to share evidence-based information, and to provide support and BF skill development. The inclusion of positive social support networks was also emphasized as an important component of BF support. It is evident that comprehensive BF support is required, particularly for populations with low rates of BF, such as adolescents. This BF support includes social support and knowledge, which are two components of our BF program.

BF in public emerged as a concern for the adolescents, which was frequently discussed in the BF classes and peer-support sessions. Mixed public attitudes regarding BF in public have been documented, with only 75% of the public in Ottawa, Ontario supporting BF in restaurants and malls despite women’s legal right to breastfeed anytime, anywhere [26]. As such, the youth required concrete strategies to address unsolicited comments received from the members of the public. This finding of unwanted remarks was reported in a recent systematic review, which was noted to affect the BF methods of adolescents [27]. Further work was done at the outreach centre to navigate the stigma of BF in public and empower youth to feel confident in their feeding choices [28]. Although not directly evaluated as part of the pilot study, the inclusive staff training, designed to reach all staff regardless of role, helped foster consistent knowledge and attitudes toward BF across the organization and helped to address stigma at the local level.

### 4.1. Implications for Practice, Policy, and Research

Policy makers, healthcare organizations, and community service agencies must determine the ideal interventions to promote optimal population health. In the case of BF, the interventions are known. According to the results of a Cochrane Review, the characteristics of effective BF support for healthy, term mother–baby dyads include predictable, and ongoing support provided by trained individuals [29]. These characteristics enhance BF support by allowing mothers to anticipate when help will be available. Support should be tailored to the setting and the needs of the population. It is important to note that support could be provided by either professional or lay/peer supporters, or a combination of both, and that face-to-face support strategies are more likely to result in women practicing exclusive breastfeeding [29]. Our BF program shares these characteristics of effective BF support (i.e., predictable and scheduled, ongoing support). Furthermore, it is important to note that this program takes place within a one-stop, youth outreach agency, which offers a full range of programs and services such as medical care, as well as multiple social and educational programs, such as prenatal classes, parenting classes, and the new peer-support BF program [20]. Childcare and transportation assistance is available, which can significantly facilitate attendance and participation of at-risk adolescents. These attributes likely contributed to the success of this program.

The US Preventive Services Task Force, in their evidence report and systematic review on primary care intervention to support BF, confirmed that BF support interventions are associated with an increase in the rates of any BF as well as exclusive BF [30]. Recommendations from this review emphasized the importance of providing interventions during pregnancy and after birth, including the following: promoting the benefits of BF, providing practical advice and direct support on how to breastfeed, and providing psychological support. In our tailored program, the prenatal BF class addresses the first recommendation, and the peer-support group accomplishes all the recommendations.

Sipsma et al. [2], in their systematic review of BF promotion among adolescent mothers, identified that a combination of education and counseling provided by a lactation consultant–peer counselor team significantly improved both BF initiation and duration. This is consistent with systematic reviews of breastfeeding support for adult mothers [30,31]. In contrast, an updated version of the systematic review by Lumbiganon [32] reported no conclusive evidence for prenatal breastfeeding education alone; this finding of a lack of effectiveness of interventions provided solely in the prenatal period reinforces the need for multiple strategies that extend beyond pregnancy. The findings from our prospective pilot cohort study are promising, as they indicated an improved SE of adolescent mothers attending the tailored evidence-based BF program, which included prenatal education and peer support in both pregnancy and the postpartum period. These types of programs need to be adequately funded if BF initiation and continuation rates are to be improved.

Public campaigns are required to emphasize the message that breastfeeding is a protected right in Ontario and Canada. BF women of all ages would benefit from greater community support of their feeding choices and more BF-friendly businesses and public spaces in which to nourish their newborns/children without judgment or shame.

### 4.2. Strengths and Limitations

Strengths of the BF program include the provision of anticipatory guidance and timely support tailored to the education and support needs of adolescents. Our research investigated an intervention to address an identified gap according to a recent systematic review by Patil et al. [33], who identified that more research is needed to develop specific interventions for adolescent mothers, as the outcomes for this population were not as effective when compared to adult mothers. Pre- and postnatal adolescents need to understand the basic mechanisms of breastfeeding (i.e., milk production is triggered by stimulation at the breast and emptying of the breast; colostrum changes to transitional milk in the first few days and then mature milk over the first week after birth; newborns have small stomachs and therefore do not require large volumes of colostrum/milk; feedings are frequent in the early weeks after birth, as breastmilk is easily digestible, etc.).

By understanding “what is normal”, adolescents will be reassured that BF is progressing as expected, and perhaps they will not be tempted to supplement with formula. Hinic [34] reported that BSES-SF scores were greater in women with previous BF experience and lowest in women who received in-hospital supplementation. Given that most adolescents are unlikely to have BF experience, anticipatory guidance and timely support received from peers and professionals is critical [13,29]; our peer-support program effectively provides both.

We collaborated closely with the program facilitator from the youth outreach centre and public health nurses from our local public health unit who had a good rapport with pregnant and parenting clients and understood their learning and support needs. Evidence-based BF content consistent with the Baby Friendly Initiative Standards [21,22] was included in the prenatal classes and peer-support program, thereby ensuring consistency in information provided to attendees and alignment with recognized breastfeeding standards, as well as enhancing credibility and structure of the program curriculum. The use of the BSES-SF, validated for use with adolescents, to evaluate the impact of this new BF program was an additional strength of this study. The peer-mom leader training, delivered by an IBCLC and involving peer-mom leaders with at least four months of breastfeeding experience, also added to the program’s robustness by ensuring that peer support was both informed and relatable.

This study has some limitations. Firstly, our sample was small and limited to young women in Eastern Ontario who were attending a co-designed program with co-located health and social services; it is possible that adolescents in other communities do not have access to this type of wrap-around program. Although adolescents would have similar sociodemographic realities, the results may not be generalizable to other populations. Additionally, we did not assess participants’ level of family support, which may have influenced study results.

Secondly, the BF peer-support program was new, and as such, it was challenging to recruit participants and retain them in the study for four weeks. There were only 7 complete data sets, and this limited the analysis that could be completed. Adolescents are difficult to engage in services and retain as participants in research studies due to their transient nature. Further exploration of the non-respondents (post intervention) is required to determine if they had characteristics that made them different than the participants who answered both surveys. This information would potentially provide valuable insight into defining the targeted support required for adolescents who are at-risk for not BF. Lastly, it is not known if increases in BSES-SF scores were associated with improved BF outcomes such as longer BF duration, as this was not measured in the study.

## 5. Conclusions

Prenatal education and peer support adapted to the needs of adolescents appears to be associated with increased BSES-SF scores. Identifying mothers with low BF SE can enable healthcare professionals to implement targeted interventions with the goal of achieving longer BF duration rates in this at-risk population. Education and support for adolescents in the pre- and postnatal period have been reported to improve exclusive BF; however, more research is needed to develop specific interventions for adolescent mothers. This research has contributed knowledge regarding a potential intervention to improve BF rates among adolescents. Given the promising results of this prospective pilot cohort study, we conducted further research with a larger cohort to determine if participation in the tailored BF program was associated with improved outcomes such as increased rates of BF intention, initiation, and duration in adolescent mothers [35]. The results identified that BF intention and initiation rates were high, and BF duration rates appear to be longer compared with those who did not receive the intervention [35].

Importantly, due to the COVID-19 pandemic and the mandated public health measures restricting in-person contact, this program was easily modified to an online format. Program staff maintained client reach and impact despite the required transition to virtual education and support. Consistent participation was reported by the program coordinator, and new content was added to the peer-support program to meet the evolving needs of the attendees. This observation of continued high participation reinforces the adaptability and usefulness of this program in meeting the needs of young pregnant and parenting women. Future research is necessary to evaluate BF SE of current program participants who are receiving the expanded content as well as to assess the impact of implementing this program in other agencies supporting pregnant and parenting adolescents.

## Figures and Tables

**Figure 1 ijerph-22-01271-f001:**
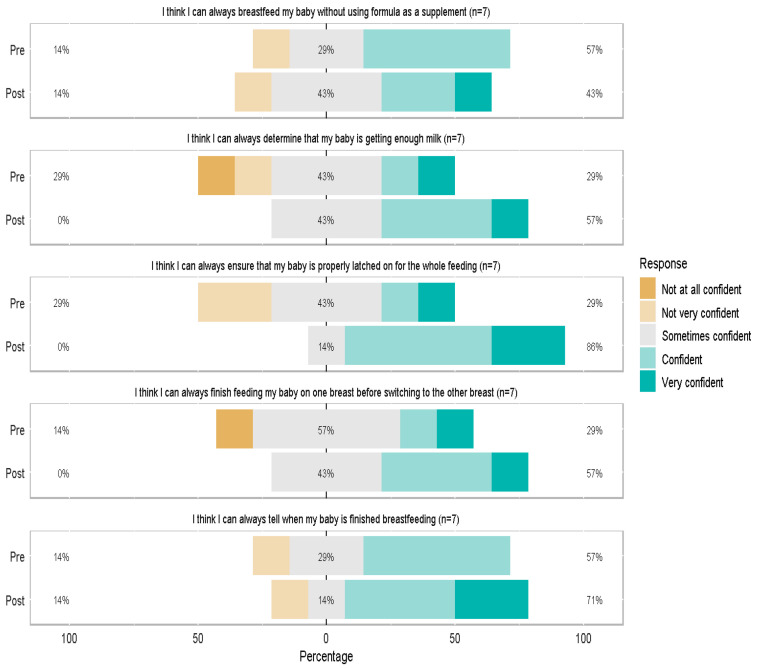
Breastfeeding Self-Efficacy Scale- Short Form items related to knowledge.

**Figure 2 ijerph-22-01271-f002:**
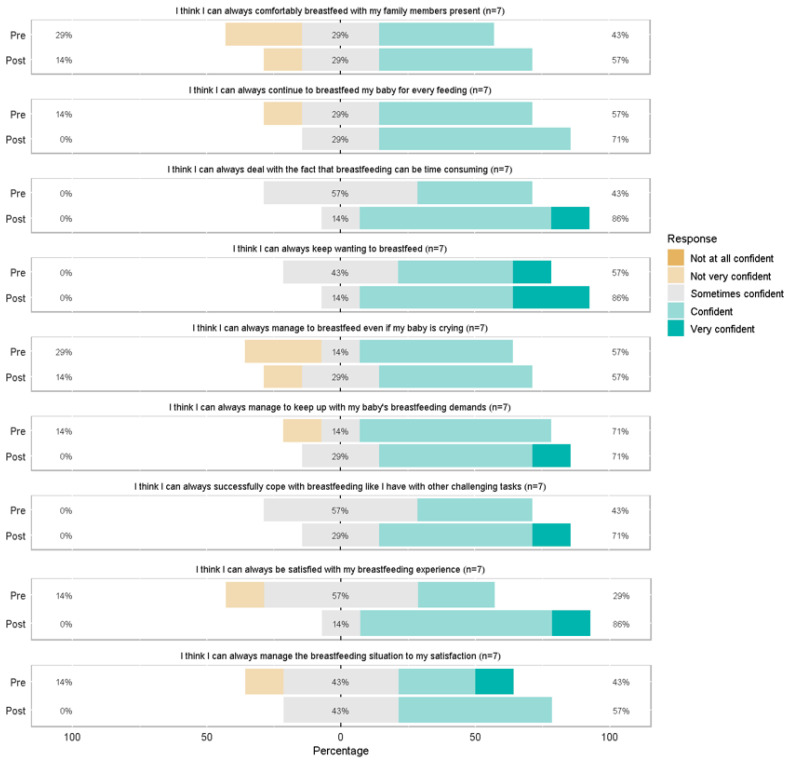
Breastfeeding Self-Efficacy Scale- Short Form items related to coping and satisfaction.

**Table 1 ijerph-22-01271-t001:** Demographics (N = 20).

Characteristic	n (%) or Mean (SD) and Range
Age (years)	16.6 (1.3) [15, 19]
Marital status	
Married	0 (0)
Single	7 (35)
Have a partner/boyfriend	12 (60)
Living with a partner/boyfriend	1 (5)
Employment	
Go to school	18 (90)
Work	1 (5)
Stay at home mom	1 (5)
Other	2 (10)
Number of pregnancies in the past ^i^	
0	11 (57.9)
1	7 (36.8)
2	1 (5.3)
Number of children ^i^	0.2 (0.4) [0, 1]
Presently pregnant (yes)	18 (90)
Previous experience breastfeeding ^i^ (yes)	2 (10.5)

^i^ n = 19, 1 record missing data.

**Table 2 ijerph-22-01271-t002:** Important concerns about breastfeeding and reasons for attending the program (N = 20).

Characteristic *	n (%)
**Important concerns about breastfeeding**	
Being unable to start breastfeeding	6 (30)
Being unable to keep breastfeeding for as long as I want	4 (20)
Feeling pressured to breastfeed	0 (0)
Not receiving support from family or friends	1 (5)
Breastfeeding in public	7 (35)
Not knowing where to go to get help with breastfeeding problems	2 (10)
Not knowing what is normal	6 (30)
Not having enough information about breastfeeding	6 (30)
Other	0 (0)
**Reason to come to the program**	
I want more information about breastfeeding.	17 (85)
I need help with my breastfeeding problems.	0 (0)
I want to meet and learn from other breastfeeding mothers.	5 (25)
I am looking for more breastfeeding resources in the community.	3 (15)
Other	2 (10)

* Participants could select more than one answer.

**Table 3 ijerph-22-01271-t003:** Pre vs. post overall BSES-SF score in all those who have complete data (N = 7).

Descriptive Statistic *	N	Pre Score	Post Score
**Overall**	7		
Median (IQR)		50.0 (40.5, 51.5)	54.0 (51.5, 57.0)
**Pregnancy Circle prenatal class only**	5		
Median (IQR)		42.0 (39.0, 50.0)	54.0 (51.0, 59.0)
**Attended both**	1		
Median (IQR)		50.0 (50.0, 50.0)	52.0 (52.0, 52.0)

* Due to limited sample size, cannot calculate *p*-value.

## Data Availability

Data are available upon request.

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
