# Peer review of "Evaluating Breastfeeding Self-Efficacy in Adolescents Attending a Co-Designed Breastfeeding Program: A Prospective Pilot Cohort Study"

_ijerph, 2025, doi:10.3390/ijerph22081271_

Round 1
Reviewer 1 Report
Comments and Suggestions for Authors
- Currently, the introduction section of the paper is very extensive. While this was helpful and informative, to improve the flow, I would recommend that the authors make the introduction section crisp.
- Was family support assessed in the study? If it was assessed, please report on this as this will be an important covariate. If not, please discuss this as a limitation.
- Table 3: as the data is not normally distributed, please report only the median
Reviewer 2 Report
Comments and Suggestions for Authors
Overall comment: This is an important piece of research, and I appreciate the opportunity to review it. The manuscript presents a valuable evaluation of a tailored breastfeeding program for adolescent mothers. While the study was conducted in 2015, it is encouraging to see the results being disseminated now, as supporting adolescents to breastfeed successfully remains an important public health issue. If the program has continued beyond its initial pilot phase, this longevity should be clearly stated in the Introduction to help frame the rationale for the paper and reinforce its relevance today.
The study is generally clearly described, with a strong intervention design and potentially meaningful findings. To further strengthen the manuscript, I encourage the authors to clarify the scope and context of existing interventions in the Introduction, consider reframing the study as a feasibility or pilot study given the small sample size, and enhance the Discussion by explicitly referencing participant concerns (Table 2) and consolidating strengths under a dedicated Strengths and Limitations section. Additionally, the paragraph discussing the program’s adaptation during COVID-19 could be better integrated by clarifying its relevance to the current study or relocating it to the Conclusion. These revisions would improve clarity, contextual relevance, and alignment between the study’s aims, methods, and conclusions, while preserving the important contributions this research makes to adolescent maternal health.

Round 2
Reviewer 2 Report
Comments and Suggestions for Authors
It was a pleasure to read the revised manuscript, which has been considerably strengthened. I recommend it for publication and look forward to seeing it in print.